# Combined Effects of Methyldopa and Baicalein or *Scutellaria baicalensis* Roots Extract on Blood Pressure, Heart Rate, and Expression of Inflammatory and Vascular Disease-Related Factors in Spontaneously Hypertensive Pregnant Rats

**DOI:** 10.3390/ph15111342

**Published:** 2022-10-29

**Authors:** Michał Szulc, Radosław Kujawski, Przemysław Ł. Mikołajczak, Anna Bogacz, Marlena Wolek, Aleksandra Górska, Kamila Czora-Poczwardowska, Marcin Ożarowski, Agnieszka Gryszczyńska, Justyna Baraniak, Małgorzata Kania-Dobrowolska, Artur Adamczak, Ewa Iwańczyk-Skalska, Paweł P. Jagodziński, Bogusław Czerny, Adam Kamiński, Izabela Uzar, Agnieszka Seremak-Mrozikiewicz

**Affiliations:** 1Department of Pharmacology, Poznan University of Medical Sciences, Rokietnicka 3, 60-806 Poznań, Poland; 2Department of Pharmacology and Phytochemistry, Institute of Natural Fibres and Medicinal Plants—National Research Institute, Kolejowa 2, 62-064 Plewiska, Poland; 3Department of Stem Cells and Regenerative Medicine, Institute of Natural Fibres and Medicinal Plants—National Research Institute, Kolejowa 2, 62-064 Plewiska, Poland; 4Department of Biotechnology, Institute of Natural Fibres and Medicinal Plants—National Research Institute, Wojska Polskiego 71b, 60-630 Poznań, Poland; 5Department of Breeding and Botany of Useful Plants, Institute of Natural Fibres and Medicinal Plants—National Research Institute, Kolejowa 2, 62-064 Plewiska, Poland; 6Department of Biochemistry and Molecular Biology, Poznan University of Medical Sciences, Święcickiego 6, 60-781 Poznań, Poland; 7Department of General Pharmacology and Pharmacoeconomics, Pomeranian Medical University in Szczecin, Żołnierska 48, 70-204 Szczecin, Poland; 8Department of Pediatric Orthopedics and Traumatology, Pomeranian Medical University, Unii Lubelskiej 1, 71-252 Szczecin, Poland; 9Division of Perinatology and Women’s Diseases, Poznan University of Medical Sciences, Polna 33, 60-535 Poznań, Poland

**Keywords:** placenta, *Scutellaria baicalensis*, baicalein, methyldopa, blood pressure, mRNA expression, inflammatory and vascular factors, oxidative stress

## Abstract

The aim of the study was to investigate the effect of baicalein or *Scutellaria baicalensis* root extract interaction with methyldopa in pregnant spontaneously hypertensive rats (SHR) at the pharmacodynamic, molecular, and biochemical levels. The rats, after confirming pregnancy, received baicalein (200 mg/kg/day, p.o.) and extract (1000 mg/kg/day, p.o.), in combination with methyldopa (400 mg/kg/day; p.o.), for 14 consecutive days, 1 h before blood pressure and heart rate measurements. In the heart and placenta from mothers after giving birth to their offspring, mRNA expression of factors related to inflammatory processes (TNF-α, Il-1β, IL-6) and vascular diseases (TGF-β, HIF-1α, VEGF, PlGF) was measured. Levels of markers of oxidative stress (superoxide dismutase and malondialdehyde) in the placenta and indicators of myocardial damage (troponin cTnC and cTnI, creatine kinase, myoglobin, and lactate dehydrogenase) in the heart were also assessed. Baicalein co-administered with methyldopa was associated with reduced blood pressure, especially during the first three days. The interactions were more pronounced for such factors as TGF-β, HIF-1α, VEGF, and PlGF than TNF-α, Il-1β, and IL-6. Combined application of baicalein and extract with methyldopa may be of value in the development of a new antihypertensive medication intended for patients suffering from preeclampsia or pregnancy-induced hypertension.

## 1. Introduction

Preeclampsia is a complex syndrome specific to pregnancy [1]. The disease is characterized by edema, hypertension (≥140/90 mm Hg), and proteinuria (≥300 mg in 24-h collection of urine) and it affects 3–4% of all pregnancies worldwide [2]. It is a condition that involves numerous and constant interactions among the placental, immunologic, and cardiovascular systems [3]. Risks to the fetus due to its occurrence and progression include premature delivery, growth retardation, and death. Treatment of severe hypertension is necessary to prevent cerebrovascular, cardiac, and renal complications in the mother [4]. 

Drug treatment options in preeclampsia are limited because of the tendency of some antihypertensive drugs, such as angiotensin-converting enzyme (ACE) inhibitors and angiotensin II receptor (AT1) antagonists (sartans), to cause teratogenic effects on the fetus [5]. Hence, so far, methyldopa is one of the appropriate first-line agents recognized as safe for the fetus [4]. Methyldopa is a drug belonging to a group of drugs acting on the central nervous system to reduce cardiovascular system tension [6,7,8]. Although its detailed mechanism of action is not fully understood, it is considered to be an agonist at peripheral and central presynaptic α2-adrenergic receptors. The peripheral action is based on the stimulation of presynaptic inhibitors of the α2 receptors, thereby inhibiting the release of norepinephrine from the neuron, and consequently causing relaxation of blood vessels [7]. However, the mechanism of this action is based on the fact that methyldopa, entering the central nervous system, acts through its active metabolite, i.e., methyl norepinephrine (displacing available norepinephrine, noradrenaline, by competition), stimulates presynaptic α2-adrenergic receptors within the nucleus of the solitary pathway, which inhibits sympathetic neurons that innervate the heart muscle and blood vessels, and stimulates the vagus nerve, resulting in an increase in parasympathicotonia. As a result, blood pressure is lowered and peripheral resistance is reduced [9]. The mechanism of action of methyldopa also consists in inhibiting the activity of aromatic L-amino acid decarboxylase in (DOPA–decarboxylase)—an enzyme responsible for synthesis of dopamine and serotonin. Thus, the involvement of DOPA–decarboxylase in the metabolism of methyldopa leads to impairment of the process of L-DOPA conversion to noradrenaline catalyzed by this enzyme [6]. However, treatment of preeclampsia remains sometimes challenging due to some adverse side effects occurring as a result of taking this drug, such as hepatotoxicity [10], and it may be hard to tolerate due to causing dizziness, depression or headache [7,11].

One possibility of potentiating antihypertensive effects with a possible reduction of side effects is combined therapy, which is nowadays standard in treating non-pregnant hypertension [12]. This kind of therapy shows a greater chance of obtaining a hypotensive response in a pathogenically complex disease such as preeclampsia due to different mechanisms of drug action, greater potency of obtaining the hypotensive effect due to synergistic action, and the possibility of using drugs in doses that give the lowest possible likelihood of adverse effects. Therefore, searching for new drugs for combined therapy is necessary. One source of potential new drugs is herbal plants, including known plant materials with proven antihypertensive effects [13].

Baikal skullcap (*Scutellaria baicalensis* Georgi, Lamiaceae) is known as one of the most popular medicinal plants and is used in several countries. Roots of *S. baicalensis* possess a long tradition of usage in Chinese medicine and a well-known long history of usage as herbal medicine. The roots of this plant have been used to treat many ailments relating to inflammation [14], and exert hepatoprotective [15], cardiovascular, neuroprotective, antioxidant, and antiviral activity [16]. Preparations based on this plant material have been used as a treatment for diarrhea, dysentery, hypertension [15,16], cancer, and influenza [14] as well. Moreover, dried parts of *S. baicalensis* in China and Japan have been described as having curative effects on hepatitis, cirrhosis, jaundice, hepatoma, leukemia, hyperlipemia, atherosclerosis, and inflammation [17]. Recently, *Scutellariae radix* extract has been shown to inhibit replication of the SARS-CoV-2 virus [18]. What is especially interesting, scientific literature also indicates that Chinese herbal medicine is still used as a natural and safe way to treat diseases specific to pregnancy. However, it is always an alternative treatment to conventional medicine for various diseases of pregnant women, such as vomiting, pregnancy-induced hypertension, or miscarriage—*Scutellariae radix* (Chinese Huang Qin) is one of the three herbs most frequently prescribed for and used by pregnant women in Taiwan [19].

Flavonoids are a major group of plant-derived compounds with polyphenolic structure, with a vital role as important bioactive components in many plants, including Baikal skullcap. The main constituents of the *Scutellaria* root-specific 40-deoxyflavones are baicalein, baicalin, wogonin, wogonoside, and oroxylin A [14,20,21,22]. It is known that flavonoids, as the main active substances of *S. baicalensis*, are likely to act directly on immune cells (mostly lymphocytes, macrophages, dendritic cells, monocytes, and neutrophils) [22]. They act by inhibiting production of inflammatory cytokines and other inflammatory mediators, including prostaglandins, leukotrienes, and reactive oxygen species.

In recent years, there has been growing evidence that quercetin, a flavonoid well known for its antihypertensive action, may be considered a source of even a prototype for a safe antihypertensive drug [23]. Also, other flavonoids have also been considered within a similar field of pharmacological applicability, such as apigenin [24], chrysin [25], baicalein [26,27], and scutellarin [28]; all exhibit vasoprotective properties, and many other activities, such as anti-oxidation via several pathways, anti-inflammation, anti-ischemia, cardioprotection, and anti-hypertension [29]. These flavonoids have not demonstrated teratogenic or abortive effects, so they are generally considered safe [30,31].

Since it is known that hypertension is very often associated with damage to the vascular endothelium [32], the results of the experiments on the protective effects of baicalin from *S. baicalensis* on vascular endothelial cells are intriguing [33]. This is evident, for example, in the case of preclinical studies showing the ability of baicalein to alleviate symptoms of pregnancy-induced hypertension, as well as vascular endothelial and placental injury [33]; or in the case of acute liver and kidney injury in a rat preeclampsia model when baicalein dose-dependently reduced blood pressure and apoptosis of kidney and liver cells [34]. 

In our previous study, we investigated the effects of the combined administration of apigenin, baicalein, chrysin, quercetin, and scutellarin with methyldopa on the expression of selected markers of inflammation (tumor necrosis factor α (TNFα); interleukin 1β (IL-1β); interleukin 6 (IL-6)) and vascular effects (hypoxia-inducible factor 1α (HIF-1α); placental growth factor (PlGF); transforming growth factor β (TGF-β); vascular endothelial growth factor (VEGF)), at the mRNA and protein levels, in human trophoblast-origin cells (JEG-3 cells) and human umbilical vein endothelial cells (HUVEC) [35]. We found that every combined administration of a flavonoid and methyldopa in these cells induced a down-regulating effect on all investigated factors, except PlGF, especially at the mRNA level. It is known that hypertension generally raises expression of TNF-α, IL-1β, IL-6, HIF-1α, TGF-β, and VEGF at mRNA and/or protein levels, so the results obtained in the studied model may provide a positive prognostic factor for such activity in vivo [35]. 

Based on the above, in this study, we decided to evaluate the in vivo antihypertensive effect of combined administration of flavonoid with methyldopa, in pregnant spontaneously hypertensive rats (SHR) as a model. Baicalein in combination with methyldopa was selected for the study because the results obtained previously were the strongest for this flavonoid [35]. The effects of such a combination were analyzed not only for the presumed antihypertensive effect but also for the potential anti-inflammatory or circulatory effect in the collected biological material (heart, placenta) from mothers after giving birth to their offspring. The molecular analyses were focused on evaluation of expression changes of above-mentioned factors related to inflammatory processes (TNF-α, Il-1β, IL-6), vascular diseases (TGF-β, HIF-1α, VEGF, PlGF), and representatives of the oxidative stress-related enzymes superoxide dismutase (SOD) and malondialdehyde (MDA/TBARS), together with the quantification of selected indicators of myocardial damage (troponin cTnC and cTnI, creatine kinase, myoglobin, and lactate dehydrogenase) in the heart tissue collected from the above-mentioned rats.

## 2. Results 

### 2.1. Analysis of Bioactive Compounds in the Extract 

The chromatographic tests carried out on the extract gave the following results: baicalin—13.6 ± 0.1 g/100 g, baicalein—86.8 ± 4.4 mg/100 g, wogonoside—1.41 ± 0.04 g/100 g, chrysin 7-glucuronide—0.47 g/100 g ± 0.02 g/100 g. This is consistent with the results of other authors who obtained similar results for the aqueous extract of *S. baicalensis* roots [36].

### 2.2. Cardiovascular Measurements

Measurements of systolic blood pressure (SBP), diastolic blood pressure (DBP), and heart rate (HR) were performed on each of the 13 days of the experiment in the morning, noon, and evening hours. Graphical presentation of the results was provided for clarity at the 1st, 3rd, 7th, 10th, and 13th day of the experiment.

#### 2.2.1. Morning Values of SBP, DBP, and HR

In the studies on the influences of the tested compounds on SBP, DBP, and HR measured in the morning, statistically significant overall variability was observed (for SBP (ANOVA II—total effect: F(4, 46) = 153.8, *p* < 0.0001; time effect: F(12, 552) = 47.3, *p* < 0.0001; and interaction: F(48, 552) = 18.3, *p* < 0.0001), for DBP (ANOVA II—total effect: F(4, 46) = 50.6, *p* < 0.0001; time effect: F(12, 552) = 7.79, *p* < 0.0001; and interaction: F(48, 552) = 5.86, *p* < 0.0001) and for HR (ANOVA II—total effect: F(4, 46) = 6.96, *p* < 0.001; time effect: F(12, 552) = 5.59, *p* < 0.0001; and interaction: F(48, 552) = 3.79, *p* < 0.0001)). During the entire experiment, the SHR control rats had significantly higher SBP, DBP, and HR than the WKY control rats (*p* < 0.05) and the cardiovascular parameters practically did not change in the following days of the experiment (Figure 1, Figure 2 and Figure 3). 

Administration of methyldopa significantly (*p* < 0.05) reduced SBP compared to SHR control, whereas DBP and HR were significantly reduced by the drug (*p* < 0.05) in SHR animals only on day three, while at the remaining time points the values did not differ significantly from the SHR control group (Figure 1, Figure 2 and Figure 3). 

Administration of baicalein together with methyldopa significantly (*p* < 0.05) reduced SBP compared to the methyldopa group on days 3 (the strongest effect) and 7, but for DBP and HR such an effect was observed only on day 3, while on the remaining days the values remained similar to that of the SHR control group. *S. baicalensis* root extract administered together with methyldopa lowered SBP values significantly (*p* < 0.05) in relation to methyldopa on days 1 and 3 (the strongest effect). Such an effect for DBP was observed on day 3 only, but on the remaining days it was comparable to the SHR control group (especially on days 7 and 10). The effect of combined *S. baicalensis* root extract and methyldopa did not differ significantly vs. methyldopa during HR measurements.

#### 2.2.2. Noon Values of SBP, DBP, and HR

In the studies on the influences of the tested compounds on the SBP, DBP, and HR measured at noon, statistically significant overall variability was observed (for SBP (ANOVA II total effect: F(4, 47) = 165.2, *p* < 0.0001; time effect: F(12, 564) = 35.9, *p* < 0.0001; and interaction: F(48, 564) = 12.5, *p* < 0.0001), for DBP (ANOVA II—total effect: F(4, 47) = 49.8, *p* < 0.0001; time effect: F(12, 564) = 14.5, *p* < 0.0001; and interaction: F(48, 564) = 10.0, *p* < 0.0001), and for HR (ANOVA II—total effect: F(4, 47) = 12.2, *p* < 0.0001; time effect: F(12, 564) = 3.60, *p* < 0.0001; and interaction: F(48, 564) = 4.14, *p* < 0.0001)). During the entire experiment, the SHR control rats had significantly higher SBP, DBP, and HR than the WKY control rats (*p* < 0.05), and the values practically did not change during the following days of the experiment (Figure 4, Figure 5 and Figure 6), although for HR on days 3 and 10 the differences did not reach significance. 

Administration of methyldopa significantly (*p* < 0.05) reduced SBP in SHR animals at all time points, with the strongest decrease on day 3, whereas during DBP and HR measurements the drug significantly (*p* < 0.05) reduced the parameters in SHR animals on days 3, 10, and 13 (for DBP) and on days 1, 3, 7, and 10 (for HR). Baicalein administered together with methyldopa significantly (*p* < 0.05) decreased SBP and DBP in relation to animals receiving methyldopa alone on days 1, 3, and 7. During HR measurements such effects were observed on days 3 and 10 only. *S. baicalensis* root extract administered together with methyldopa significantly (*p* < 0.05) lowered SBP and DBP values in relation to methyldopa on days 1, 7, 10, and 13, but during HR measurements the combined effect of *S. baicalensis* root extract and methyldopa was similar in potency to the drug alone.

#### 2.2.3. Evening Values of SBP, DBP, and HR

In the studies on the influences of the tested compounds on the SBP, DBP, and HR measured in the evening, statistically significant overall variability was observed (for SBP (ANOVA II—total effect: F(4, 47) = 112.2, *p* < 0.0001; time effect: F(12, 564) = 71.4, *p* < 0.0001; and interaction: F(48, 564) = 13.9, *p* < 0.0001), for DBP (ANOVA II—total effect: F(4, 47) = 39.7, *p* < 0.0001; time effect: F(12, 564) = 21.3, *p* < 0.0001; and interaction: F(48, 564) = 7.34, *p* < 0.0001) and for HR (ANOVA II—total effect: F(4, 47) = 13.2, *p* < 0.0001; time effect: F(12, 564) = 11.3, *p* < 0.0001; and interaction: F(48, 564) = 5.16, *p* < 0.0001)). During the entire experiment, the SHR control rats had significantly higher SBP, DBP, and HR than the WKY control rats (*p* < 0.05), and the values practically did not change in the following days of the experiment (Figure 7, Figure 8 and Figure 9). 

Administration of methyldopa significantly (*p* < 0.05) reduced SBP in SHR animals at all time points, with the strongest decrease on day 3, whereas the drug significantly (*p* < 0.05) reduced DBP and HR in SHR animals on days 1 and 3 only, while at the remaining time points the obtained values did not differ significantly from the SHR control group. Baicalein and *S. baicalensis* root extract administered together with methyldopa significantly (*p* < 0.05) decreased SBP in relation to animals receiving methyldopa alone on days 1, 3, and 7 (on this day baicalein only), whereas the flavonoid and *S. baicalensis* root extract when administered together with methyldopa significantly (*p* < 0.05) decreased DBP and HR in relation to animals receiving methyldopa alone on day 3 only (the extract also on day 1). Surprisingly, on day 13 the extract was associated with an elevation of HR when compared both with SHR control and methyldopa groups.

#### 2.2.4. Comparative Analysis of the Influence of Time of Day and Duration of the Experiment

In the next stage of the analysis, changes in SBP, DBP, and HR were compared between administered methyldopa and its combination with baicalein and *S. baicalensis* root extract with regard to the time of day of administration. Figure 10, Figure 11 and Figure 12 show the mean SBP, DBP, and HR over a period of 13 days of the experiment, respectively. The main effects (group effects) were found to be statistically significant (for SBP: ANOVA II: F(4, 48) = 15.4, *p* < 0.0001; for DBP: ANOVA II: F(4, 60) = 11.1, *p* < 0.0001; and for HR: ANOVA II: F(4, 60) = 11.1, *p* < 0.0001). 

However, further analysis showed that no effect of the day of administration was noted in the investigated groups (for SBP: ANOVA II: F(2, 120) = 0.01, *p* > 0.05; for DBP: ANOVA II: F(2, 120) = 0.448, *p* > 0.05; and for HR: ANOVA II: F(2, 120) = 0.441, *p* > 0.05). It was also found that administration of methyldopa as well as the combination of the compound or the extract with methyldopa significantly reduced SBP and DBP, and these effects were exacerbated by administration of methyldopa with baicalein in relation to methyldopa alone (*p* < 0.05). During HR measurements, the administration of methyldopa as well as the combination of the compound (but not the extract) with methyldopa significantly reduced the parameter. These effects were not affected by administration of methyldopa with the extract in relation to methyldopa alone (*p* > 0.05).

### 2.3. Body Mass

The body mass of the animals was measured at the beginning of substance administration and on the last day of the experiment. During the entire experiment, statistically significant overall variability was observed (ANOVA II—total effect: F(4, 47) = 41.7, *p* < 0.0001; time effect: F(1, 47) = 622.0, *p* < 0.0001; and interaction: F(4, 47) = 44.1, *p* < 0.0001). It was noted that at the beginning of the experiment all SHR rats had significantly lower (*p* < 0.05) body masses than the WKY control rats (Figure 13). At the end of the experiment, the weight of the animals increased significantly in all groups (*p* < 0.05), although the SHR control rats still weighed less than the WKY control, and the administration of all substances significantly inhibited the increase in body mass of the animals, although the type of substance administered was not a differentiating factor. 

### 2.4. VEGF

Based on the obtained data on VEGF mRNA expression in the heart, there was significant variability between the groups (ANOVA I: F(4, 47) = 8.14, *p* < 0.0001). Further statistical analysis showed that SHR control rats had lower values by 7% vs. the WKY control group, whereas administration of methyldopa did not affect the parameter (insignificantly decreased) vs. the SHR control (Table 1). The extract when administered together with methyldopa also lowered VEGF mRNA values vs. methyldopa (by 6%, respectively). However, the baicalein and methyldopa coadministration led to an increase of the values almost to the values observed in the WKY control group. 

Analyzing the results for VEGF mRNA expression in the placenta, there was significant variability between the groups (ANOVA I: F(4, 28) = 12.3, *p* < 0.0001). Further statistical analysis showed that SHR control rats had higher mRNA values (by 35%) vs. WKY control group (*p* < 0.05) (Table 1). Methyldopa administration significantly lowered the parameter vs. SHR control animals (by 17%), similar values were obtained after combined extract and methyldopa administration, while baicalein when administered together with methyldopa significantly increased mRNA expression (by 17%) to the values observed for SHR control rats. It can be seen that both methyldopa itself and its administration together with the extract led to mRNA expression similar to that in WKY control animals.

### 2.5. HIF-1α

On the basis of the obtained results of HIF-1α mRNA expression in the aorta, statistically significant variability between the groups was observed (ANOVA I: F(4,47) = 3.09, *p* < 0.05). Further statistical analysis showed that SHR control rats had lower values by 9% vs. WKY control group, whereas administration of methyldopa did not affect the parameter vs. SHR control (Table 1). However, when baicalein or the extract was treated with methyldopa together, there were no differences between their HIF-1α mRNA expression levels and those of the methyldopa group. 

Analyzing the results for HIF-1α mRNA in the placenta, statistically significant variability between the groups was observed (ANOVA I: F(4, 28) = 7.80, *p* < 0.001). 

Further statistical analysis showed that SHR control rats had higher mRNA values (by 34%) vs. the WKY control group (*p* < 0.05) (Table 1). Methyldopa administration significantly lowered the parameter vs. SHR control animals (by 23%), similar values were obtained after combined extract and methyldopa administration, while baicalein when administered together with methyldopa significantly increased mRNA expression (by 15%) to the values observed for SHR control rats. It can be seen that both methyldopa itself and its administration together with the extract led to HIF-1α mRNA expression similar to that in WKY control animals.

### 2.6. TGF-β

Based on the obtained data on TGF-β mRNA expression in the heart, there was significant variability between the groups (ANOVA I: F(4, 47) = 8.92, *p* < 0.0001). Further statistical analysis showed that SHR control rats had 12% lower values compared to the WKY control group, whereas administration of methyldopa did not affect the parameter vs. SHR control (Table 1). However, when baicalein or the extract was administered with methyldopa together, there were no differences in the TGF-β mRNA expression levels compared with the methyldopa group. 

On the basis of the obtained results of TGF-β mRNA in the placenta, there was significant variability between the groups (ANOVA I: F(4, 28) = 5.19, *p* < 0.01). Further statistical analysis showed that SHR control rats had higher mRNA values (by 18%) vs. WKY control group (*p* < 0.05) (Table 1). Methyldopa administration significantly lowered the parameter vs. SHR control animals (by 13%) leading to control values in WKY rats. In contrast, the combined baicalein treatment with methyldopa significantly increased the mRNA values vs. methyldopa (by 20%) leading to SHR control values, whereas the combined extract with methyldopa did not significantly change the TGF-β mRNA expression observed in the methyldopa group. 

### 2.7. PlGF

Based on the obtained data on PlGF mRNA expression in the heart, statistically significant variability between the groups was observed (ANOVA I: F(4, 46) = 4.89, *p* < 0.01). It was found that SHR control animals had statistically significantly higher (*p* < 0.05) mRNA expression values when compared with WKY control rats (by 10%) (Table 1). In contrast, the combined baicalein or the extract treatment with methyldopa significantly increased the mRNA values vs. methyldopa (by 12%), leading to SHR control values. 

Analyzing the results for PlGF mRNA in the placenta, there was no statistically significant variability between the groups (ANOVA I: F(4,28) = 1.10, *p* > 0.05), and all expression values of the studied groups were at a similar level (Table 1).

### 2.8. TNF-α

On the basis of the obtained results of TNF-α mRNA expression in all studied tissues, it was found that there was no statistically significant variability between the groups (heart ANOVA I: F(4, 47) = 0.91, *p* > 0.05, placenta ANOVA I: F(4, 28) = 1.27, *p* > 0.05). To sum up, in the experimental system, neither the administration of methyldopa nor its interaction with the flavonoids or extract changed this parameter.

### 2.9. IL-1β

Based on the obtained data on IL-1β mRNA expression in the heart, statistically significant variability between the groups was observed (ANOVA I: F(4, 47) = 10.0, *p* < 0.0001). It was found that SHR control animals had statistically significantly lower (*p* < 0.05) mRNA expression values when compared with WKY control rats (by 8%) (Table 1). Methyldopa administration significantly (*p* < 0.05) lowered the parameter when compared with SHR control animals (by 10%). In contrast, the combined baicalein treatment with methyldopa significantly increased the mRNA values vs. methyldopa (by 14%), leading to SHR control values. Methyldopa with the extract had a similar effect when compared with the methyldopa, but the difference was not statistically significant (*p* > 0.05).

Analyzing the results for IL-1β mRNA expression in the placenta, it was found that there was no statistically significant variability between the groups (ANOVA I: F(4, 28) = 0.972, *p* > 0.05). To sum up, in the experimental system, neither the administration of methyldopa nor its interaction with baicalein or the extract changed this parameter (Table 1). 

### 2.10. IL-6

Based on the obtained data on IL-6 mRNA expression in the heart, statistically significant variability between the groups was observed (ANOVA I: F(4, 47) = 2.50, *p* = 0.05). Further statistical analysis showed that SHR control rats did not statistically differ in this parameter when compared with WKY control animals (Table 1). 

Similarly, methyldopa administration did not change the mRNA expression vs. SHR control animals. Also, the combined baicalein or the extract treatment with methyldopa did not differ statistically in the mRNA values vs. methyldopa.

Analyzing the results for IL-6 mRNA expression in the placenta, no statistically significant variability was found between the groups (ANOVA I: F(4, 28) = 0.45, *p* > 0.05). To sum up, in the experimental system, neither the administration of methyldopa nor its interaction with baicalein or the extract changed this parameter (Table 1). 

### 2.11. Myocardial Proteins 

Results of the effect of methyldopa and its combination with the flavonoid or extract from *S. baicalensis* roots on levels of selected proteins related to heart damage parameters (creatine kinases (B and M), myoglobin, troponins (cTnT and cTnI), and lactate dehydrogenase (A)) in SHR pregnant rats are shown in Table 2.

#### 2.11.1. Creatine Kinase B-Type (CKB)

Based on the obtained data of CKB protein level, statistically significant variability between the groups was observed (ANOVA I: F(4, 43) = 6.12, *p* < 0.001). Further statistical analysis showed that SHR control rats did not statistically differ in this parameter when compared with WKY control animals (*p* > 0.05). Methyldopa administration significantly (*p* < 0.05) decreased CKB vs. SHR group (by 24%). Baicalein when administered together with methyldopa decreased CKB values significantly (*p* < 0.05) when compared with the methyldopa group (by 24%). The effect of the extract treatment with methyldopa was completely different, since it led to an increase of CKB to values which were significantly higher than in the methyldopa group but also much higher than those observed in WKY animals (by 26%).

#### 2.11.2. Creatine Kinase M-Type (CKM)

On the basis of the obtained results of CKM protein level, statistically significant variability between the groups was observed (ANOVA I: F(4, 42) = 6.05, *p* < 0.01). During the entire experiment, the SHR control rats did not differ significantly when compared with WKY control rats (*p* > 0.05). CKM levels after administration of methyldopa alone and its combination with baicalein were similar to SHR rats and did not differ significantly. Only the administration of the extract together with methyldopa significantly (*p* < 0.05) increased the value of this parameter when compared with methyldopa group (by 82%) to levels of WKY that, but not significantly (*p* > 0.05), exceeded the values for the WKY group.

#### 2.11.3. Myoglobin

Analyzing the results for myoglobin levels, statistically significant variability between the groups was observed (ANOVA F(4,39) = 3.95, *p* < 0.01). Further statistical analysis showed that in SHR control rats this parameter increased significantly (by 59%) when compared with WKY control animals (*p* < 0.05). Methyldopa administration significantly (*p* < 0.05) decreased myoglobin level compared to the SHR group (by 77%, clearly below the value for the WKY group). After administration of methyldopa combination with baicalein, the results were similar to the value of methyldopa alone and did not differ significantly. Only the administration of the extract together with methyldopa significantly (*p* < 0.05) increased the value of this parameter when compared with the methyldopa group (by 82%) to values comparable for WKY.

#### 2.11.4. Troponin T (cTnT)

On the basis of the obtained results of cTnT level, insignificant variability between the groups was observed (ANOVA F(4,42) = 2.52, *p* < 0.06). It was found that the SHR control rats did not differ significantly when compared with WKY control rats (*p* > 0.05). cTnT level after administration of methyldopa alone was lower vs. the SHR group (*p* < 0.05), whereas its combination with baicalein or the extract did not differ significantly when compared with the methyldopa group and the values were similar to SHR rats. 

#### 2.11.5. Troponin I (cTnI)

Based on the obtained data of cTnI level, statistically significant variability between the groups was observed (ANOVA F(4, 45) = 3.44, *p* < 0.05). Further statistical analysis showed that SHR control rats did not differ when compared with WKY animals (*p* > 0.05). Methyldopa administration significantly (*p* < 0.05) decreased the level vs. the SHR group (by 37%). Baicalein and the extract after their combined administration with methyldopa resulted in the same changes, but was not significant vs. methyldopa alone. 

#### 2.11.6. Lactate Dehydrogenase A (LDH-A)

Analyzing the results for LDH-A levels, significant variability between the groups was observed (ANOVA F(4,42) = 3.89, *p* < 0.05). It was found that the SHR control rats did not differ significantly when compared with WKY rats (*p* > 0.05). LDH-A level after administration of methyldopa alone was lower vs. SHR control rats (*p* < 0.05). The combined baicalein and the extract treatment with methyldopa did not significantly increase the parameter in comparison to the methyldopa group.

### 2.12. Factors Related to Oxidative Stress 

#### 2.12.1. Malonyldialdehyde Concentration (MDA)

On the basis of the obtained results of MDA levels in the placenta, statistically significant variability between the groups was observed (ANOVA I: F(4,41) = 4.91, *p* < 0.05). 

It was found that SHR control animals had higher MDA values when compared with WKY control rats (by 25%, *p* < 0.05) (Table 3). Methyldopa administration decreased the parameter in comparison to the SHR control group (by 49%). In contrast, the combined baicalein and extract treatment with methyldopa increased the MDA level vs. methyldopa (by 86% and 48%, respectively), leading to values as much as 50% higher (in the case of baicalein), while for the extract these values were similar to that of the WKY control group. 

#### 2.12.2. Activity of Superoxide Dismutase (SOD)

Based on the obtained data on activity of SOD in the placenta, statistically significant variability between the groups was observed (ANOVA I: F(4,35) = 5.99, *p* < 0.01). It was found that SHR control animals had statistically significant (*p* < 0.05) higher SOD activity values when compared with WKY control rats (by 51%) (Table 3). Methyldopa administration significantly (*p* < 0.05) decreased SOD values when compared with SHR control animals (by 23%). In contrast, the combined baicalein treatment with methyldopa significantly increased SOD activities vs. methyldopa (by 24%), leading to SHR control values. However, the combined extract and methyldopa administration did not change the parameter in comparison to the methyldopa group, leading to values observed in the WKY control group.

## 3. Discussion

The studies presented in this paper were carried out on SHR rats as a model of arterial hypertension. These rats showed significantly higher SBP and DBP throughout the experiment when compared with WKY animals, which is consistent with the phenotypic feature of the animals used [37], although it is known that pregnancy somewhat lowers these parameters in such animals [38]. While there are many other models that are used to mimic preeclampsia in animals [39], due to the presently described complexity of this disease at different levels of analysis using various omics technology platforms, including epigenetics, genome-wide association studies, transcriptomics, proteomics, and metabolomics, none of them are perfect [40,41], and the phenotypically observed feature of increased arterial hypertension in SHR rats allowed us to study one of the basic features of this disease in model conditions. The SHR rats also showed an accelerated HR in comparison to WKY animals, which is consistent with the opinion that this parameter is highly predictive of hypertension occurrence in the animal population studied [42].

In this study, the administration of methyldopa at a dose of 400 mg/kg/day was used, similarly to that of Podjarny et al. who investigated the effect of methyldopa on renal function in rats with L-NAME-induced hypertension in pregnancy [43]. However, the dose of flavonoid (200 mg/kg/day) was used on the basis of baicalein studies in which it showed a protective effect against hypertension associated with diabetes [44] and the results in which it attenuated metabolic disease in fructose-fed rats [45]. The doses used in pharmacological studies of *S. baicalensis* root extracts vary considerably and are used in the range of 0.1–1.0 g/kg (p.o.) in in vivo studies [46]. The extracts are generally safe, since in the toxicological studies of ethanol extracts of this plant, only after the administration of 2500 mg/kg p.o. did the liver tissue of the rats show some reversible inflammatory changes [45]. Taking into account the physiological condition of the animals, a dose of 1000 mg/kg/day was selected.

The strongest antihypertensive effect was found for measurements taken at noon, where for the averaged measurement of SBP at that time of day, it was noted that SHR had hypertension (SHR—182 mmHg vs. 140 mmHg for WKY), and methyldopa lowered systolic blood pressure (to 130 mmHg), especially on the third day of the experiment. Similar effects were found for DBP (SHR—108 mmHg vs. 85 mmHg for WKY) and methyldopa alone lowered systolic blood pressure in SHR rats (to 97 mmHg). The effects of methyldopa are consistent with the results obtained by other authors in SHR animals [47,48]. Similar effects were observed in patients with gestational hypertension and preeclampsia [49]. The described action, which occurred most strongly at noon, was probably related to the experimental system. All substances were administered in the morning and evening and the effect was measured after 1 h from administration, while no test compounds were administered at noon, but the antihypertensive effect was nevertheless the strongest, probably related to pharmacokinetics. On the other hand, the combined administration of baicalein with methyldopa with this drug caused in SHR a very strong and significant reduction in SBP pressure, much stronger than the effect of methyldopa alone. These effects were present throughout the study, but in the first 3 days they were the strongest, because baicalein in combination with methyldopa had a stronger effect (SBP decrease on day 3 to 110 mmHg) than methyldopa (which caused e.g., on day 3 a decrease to 122 mmHg), although the effects were also pronounced for the remaining days. It should be emphasized that the effect of the methyldopa extract was particularly noticeable on the last two days of measurement.

Similarly, the effect of baicalein administered with methyldopa on DBP in SHR rats was very strong (decrease to 92 mmHg), much stronger than methyldopa alone (decrease to 97 mmHg), In the second week of the study, rats receiving methyldopa + baicalein had the same DBP as control SHR rats (e.g., day 13—110 mmHg); in other words, the action of the combination of methyldopa and the flavonoid was not so effective. On the other hand, the combination of methyldopa and the extract showed a clearly stronger effect on the 7^th^, 10^th^, and 13^th^ day of the experiment. It would follow that the combined administration of the tested flavonoid with methyldopa is effective only during the first days of use, while the effect of the extract with methyldopa appears later. The cause of such an effect is difficult to determine at the present stage of research. It should be noted, however, that when the mean effect of the days of the experiment is averaged, in all cases, the values of the control animals SHR, WKY, after administration of methyldopa or the combination of flavonoid or extract + methyldopa (for SBP or DBP) do not differ significantly between administration in the morning, at noon, and in the evening in individual groups. The observed result of the interaction and the stronger effect of methyldopa after combined administration with baicalein may be due to the anti-hypertensive properties of the flavonoid. It is known from the studies performed by Ding et al. [20] that baicalin (in the form of baicalein glycoside) lowered the blood pressure in SHR rats, decreased vascular tension in SHR aortas, and attenuated phenylephrine action as well as angiotensin II (Ang II) and potassium chloride (KCl)-induced vasoconstriction in SHR aortas. Another similar study published by Liu et al. [18] documented that baicalin pretreatment in rats alleviated the Ang II induced constriction of the abdominal aortic ring, while it promoted NE pre-contracted vasodilation of the abdominal aortic ring, at least partly dependent on the L-type calcium channel. They concluded that baicalin acts as a blood pressure lowering agent [18]. It appears that this baicalin profile can be comparable to that of baicalin. It is known that aglycones (here baicalein) are absorbed into the epithelium of the small intestine by passive diffusion. On the other hand, glycosides (here baicalin), to be absorbed, first must be converted to aglycons, which is achieved by lactase-florin hydrolase in the lumen of the small intestine [50,51]. These observations can help better understand the potential therapeutic application of *S. baicalensis* in treatment of hypertension [20].

The strongest effect of the analyzed compound in terms of its influence on HR was found for the measurements taken at noon, as well as for the anti-hypertensive effect. Analyzing the influence of the selected flavonoid administered together with methyldopa on the mean HR values, it was found that SHR animals have a faster HR, because its mean values are 410 beats/min vs. 378 beats/min for WKY, and administration of methyldopa slowed the HR to 365 beats/min in these animals. The effect of methyldopa was in line with the observations that systemic administration of methyldopa decreased mean arterial blood pressure and caused a short lasting increase in heart rate followed by a long lasting decrease [9]. It was also noted that baicalein had a stronger inhibitory effect, especially in the first 1–3 days (e.g., on day 2 of administration in the methyldopa + baicalein group—300 beats/min, for methyldopa—330 beats/min), but in the second week these values increased and reached a similar level with the combination of methyldopa and baicalein (400 beats/min). The effect of baicalein was not so strong (only on the 3rd day), although it is known that this flavonoid has the ability to inhibit the heart rate, at the same time positively influencing the regularity of the heart beat in the diabetic cardiac autonomic neuropathy model [52]. The effect of the extract administered with methyldopa was similar to that of methyldopa alone; hence in this case it is difficult to talk about any interactions of the relevant extract with the study drug. Similar to the antihypertensive effect, it should be noted that when the mean effect of the days of the experiment on HR is averaged, in all cases, the values of the control animals SHR and WKY, after administration of methyldopa or the combination of flavonoid or extract + methyldopa HR, do not differ significantly between administration in the morning, at noon, and in the evening in individual groups.

The VEGF family, in particular, has been of great interest, due to its known association with hypertension and nephropathy, and its role as a biomarker of endothelial dysfunction, platelet activation, and tissue hypoxia [53]. VEGF is one of the essential factors maintaining physiological endothelial function in the placenta. In PE, increased placental secretion of sFLT1 inhibits VEGF and PlGF signaling in the vasculature. This causes endothelial cell dysfunction, including decreased production of prostanoids, NO, and release of prothrombotic proteins, leading to the clinical symptoms of PE [54]. In 2003–2004, several researchers reported that an abnormal increase in the level of serum sFlt-1 in pregnant mothers is well correlated with the degree of PE [55] marked mainly by hypertension and proteinuria. Those two symptoms are observed as the most common adverse effects encountered in cancer patients treated with the monoclonal antibody drug bevacizumab, a VEGF-neutralizing antibody [56]. These angiogenic factors are also potent mediators of the inflammatory response, and they augment inflammatory symptoms in patients with preeclampsia [57]. VEGF as a proangiogenic factor needs consideration as a biomarker associated with endothelial cell damage in pregnancy with severe preeclampsia [58]. However, in patients, the VEGF level in the pregnancy-induced hypertension (PIH) group was significantly lower than in the pregnancy group at advanced pregnancy, and the VEGF level significantly and gradually decreased with PIH aggravation; therefore, its role is not simple [59]. In this study, we found that in SHR rats, mRNA VEGF expression was lower in the heart, whereas in the placenta, increasing values of this parameter were found. It is in line with the results observed in pregnant women with preeclampsia, where elevation of VEGF was observed in the placenta [60]. Methyldopa lowered the expression only in the placenta in SHR rats, and this effect was similar to what was seen in our in vitro studies on JEG-3 and HUVEC cells [35]. It is also in line with the results in severe preeclampsia patients, where the drug dose dependently decreased the VEGF level [58]. However, the combination of baicalein with methyldopa significantly increased the expression in relation to the methyldopa group in all tissues. In contrast, the combined methyldopa and extract treatment led to lowering the parameter in the heart and placenta. So, it can be assumed that administering methyldopa together with flavonoids or extracts should be used with caution, because the effects of the interaction depend on the type of substance. It should also be pointed out that VEGF mRNA expression was about 50% weaker in the placenta compared with the values in the heart.

HIF-1α is a transcription factor that is regulated by hypoxia and mediates the effects of hypoxia on gene expression [61]. It is expressed in the placenta in a gestational-age dependent fashion, with levels being higher in the first trimester and declining as oxygen levels increase later in pregnancy [62]. Accumulation of HIF-1α is commonly an acute and beneficial response to hypoxia; when chronically elevated, this protein is associated with multiple disease conditions, including preeclampsia [63]. There are some data indicating that women with preeclampsia are characterized by persistently elevated placental HIF-1α levels [64,65]. In this study, we found that in SHR rats, mRNA HIF-1α expression was higher in the placenta, which corresponded to the fact mentioned above, but in the heart the opposite effects were found. Methyldopa lowered the expression in all tissues in SHR rats, which may have a positive effect. The combined treatments with methyldopa produced elevation of the expression when compared with the drug only, but values were usually below that observed for SHR control rats, which seems to be positive. It should also be pointed out that HIF-1α mRNA expression was about 50% weaker in the placenta than in the heart.

It is known that TGF-β is one of the cytokines with expression in macrophages and epithelial tissue [66], for example, in asthmatic epithelium [67], and moreover, it is associated with preeclampsia risk [62,68]. It is also well known that the increased TGF β-1 level may lead to preeclampsia [69]. In this study, we found that in SHR rats, mRNA TGF β-1 expression was higher in the placenta, which corresponded to the facts mentioned above in preeclamptic patients, whereas in the heart lowering of this parameter was found. Methyldopa lowered the expression especially in the placenta of SHR rats, which may have a positive effect. The combined baicalein and methyldopa treatment produced elevation of mRNA TGF β-1 expression especially in the placenta when compared with the drug only. However, it seems that from the practical point of view, the most favorable inhibitory interaction effect was observed after administering the extract together with methyldopa in the placenta.

PlGF is a VEGF-related molecule that is expressed at high levels by trophoblast cells in the placenta [53] and it is known to play an important role in the pathophysiology of preeclampsia [70]. It is known that the rise in plasma PlGF levels observed in normal pregnancies is significantly attenuated in pregnancies complicated by preeclampsia [71]. In this study, we found that in SHR rats, mRNA PlGF expression was higher in the heart of SHR control rats and methyldopa administration led to lowering of the expression in this tissue. The combined methyldopa and flavonoid or the extract treatment showed elevation of this parameter even to values higher than in WKY control animals, which seems to be a positive phenomenon. However, no significant effects were seen in the placenta. The effect of methyldopa on PlGF is not entirely clear, since there are even some data suggesting that methyldopa may have a specific effect on placental and/or endothelial cell function in preeclampsia patients, altering angiogenic proteins and increasing PlGF [60]. 

Detailed understanding of the pathophysiology of the inflammatory process in preeclampsia is still a subject of research [57,72]. It is known that TNF-α is a central regulator of inflammation, and this cytokine plays a crucial role in causing inflammation, predominantly by means of T lymphocytes. It is also associated with inflammatory mechanisms related to implantation, placentation, and pregnancy outcome, since over-production of TNF-α may lead to such events as recurrent pregnancy loss, early and severe preeclampsia, and recurrent implantation failure syndrome [73]. It is known that TNF-α inhibits trophoblast and endothelial cellular interactions and simultaneously decreases endothelial nitric oxide synthase (eNOS) expression, and methyldopa reversed TNF α-induced inflammation and increased eNOS expression in vitro [74]. In the present study, we observed no significant effect on the expression of mRNA TNF-α in SHR and WKY animals alone, or on the effect of methyldopa or its combined administration in all tissues. Thus, it would appear that the obtained results are inconsistent with the previously obtained effects in JEG-3 and HUVEC cells [35]. The reasons for this inconsistency are unknown. Perhaps the observed hypertension in SHR animals does not influence this parameter. This may be related to the model used, as it has been suggested that pregnant stroke-prone spontaneously hypertensive rats show higher concentrations of TNF-α compared to normal SHR [75].

It is known that IL-1 is a possible mediator of maternal endothelial dysfunction in preeclampsia [74,76], and aberrant IL-1β levels were shown to be associated with a variety of gestational diseases, such as preeclampsia, preterm labor, and spontaneous abortion [77]. It is known that IL-1 is elevated in maternal blood from women with preeclampsia [78]. In this study, we found that in SHR rats, mRNA IL-1β expression was slightly lowered in the heart of SHR control rats and methyldopa administration led to lowering of the expression in this tissue. After administration of combined methyldopa and flavonoid or the extract, this parameter increased compared with methyldopa alone, but it was not significant. However, the effect of baicalein on methyldopa combined was different, leading to values observed in WKY control rats. No significant effects were seen in the placenta. These somewhat surprising results are difficult to explain; however, in studies on women, different results were also obtained. For example, onset of labor results in elevations in amniotic fluid levels of IL-1β that are similar in preeclamptic pregnancy to those observed in normal pregnancy [79]—though this contrasts with the findings of Stallmach et al. [80]. It is proposed that these apparent contradictions between studies using immunological detection of cytokines and bioactivity studies may reflect changes in cytokine inhibitory binding proteins during preeclamptic pregnancy [62].

Numerous reports indicate that the plasma of preeclamptic patients contains elevated levels of IL-6, a multifunctional cytokine that regulates, among other things, the acute phase reaction and modulates both pro- and anti-inflammatory events, and may play roles in the pathogenesis of preeclampsia by serving as a source of a key circulating factor that promotes systemic maternal endothelial cell dysfunction [81]. Additionally, while many of the functions of IL-6 have not been explained yet, it is assumed that IL-6 is a good biomarker for adverse pregnancies [82]. In this study we found that in SHR rats mRNA IL-6 expression did not differ from WKY control rats in all tissues. Moreover, the methyldopa and its combined treatments did not change the expression both in the heart and placenta. 

Summarizing the above, the values of mRNA expression for all measured parameters in the SHR rats in the placenta decreased in relation to the heart, which have values (tissue effect) at a similar level for all parameters except PlGF and IL-6. The values specific to the placenta for SHR increase significantly, after administration of methyldopa they decrease, and the combination of methyldopa and baicalein tends to weaken the effects of methyldopa measured by mRNA expression. It seems that in the model used, the effects of methyldopa and its interaction with the selected flavonoid or the extract are more pronounced for factors related to vascular diseases in human peripheral vascular and placental endothelial cells (TGF-β, HIF-1α, VEGF, PlGF) than with inflammatory indicators (TNF-α, Il-1β, IL-6).

It is known that there are links between hypertension and myocardial infarction [7,83] and hypertensive pregnancy disorder is linked to future cardiac events since chronic hypertension is the largest contributor of all, accounting for 81% of increased cardiovascular disease risk among women who had gestational hypertension and for 48% of increased risk among women who had preeclampsia [84]. Nowadays the use of such markers as concentrations of troponin cTnC and cTnI, creatine kinase, myoglobin, and lactate dehydrogenase is commonly accepted for assessment of myocardial injury in clinical practice [85]. Therefore, we decided to measure the combined effects of methyldopa and baicalein or extract from the roots of *S. baicalensis* on levels of these myocardial injury markers in the heart of SHR pregnant rats. We found that SHR control rats generally did not differ from their WKY counterparts. This is in line with the observation that there was no difference in baseline TnI phosphorylation in SHR and WKY [86], CKB and CKM levels [87] and LDH-A [88]. We found that there was only a clear difference for myoglobin, as SHR control rats had a significantly higher level of this indicator than WKY animals. This result is probably specific because it is known that SHR rats expressed higher renal myoglobin in comparison to normal WKY rats [89]. Methyldopa administration led to a reduction of all parameters in the heart, which can probably be considered as a positive phenomenon. The combined baicalein and methyldopa administration increased myoglobin and cTnI when compared with the effect of methyldopa only, but the values were lower than in SHR control rats. A similar effect of baicalein and the drug combined treatment on cTnI was observed. In contrast, this combination led to a lower CKB level. The effect of the combined administration of methyldopa and the extract was puzzling, because while the effect expressed in the obtained values for myoglobin and cTnI was positive, the interaction of these substances led to values for CKB and CKM exceeding the measured values for the WKY control rats. The reasons for this effect of the extract administered together with methyldopa are so far difficult to explain.

Experimental evidence has shown that reactive oxygen species (ROS) play an important role in the pathophysiology of hypertension [90]. Also oxidative stress, an imbalance between free generation and antioxidant defense systems, is recognized as a key factor in the pathogenesis of many obstetrical complications [91]. Pregnancy-induced hypertension, especially preeclampsia, is a state of extremely increased oxidative stress, due to the decrease of antioxidant capacity [53]. Consequences of an increased state of oxidative stress include impairment of placental blood flow, intrauterine hypoxia of the fetus and disturbance in transfer of O_2_ [92]. It is worth remarking that preeclampsia is a multisystem disorder which involves altered homeostasis of oxidants–antioxidants, inflammatory processes, and endothelial dysfunction. Moreover, preeclamptic mothers are likely to have increased oxidative stress during pregnancy, which can adversely affect the outcome in their neonates [53,93]. We found that SHR control rats generally differed in MDA and SOD from their WKY counterparts, since the parameters were elevated in the placenta. This is in agreement with the results of others who found that SHR rats had a higher level of reactive oxygen species (ROS) determined by dihydroethidium imaging and lower antioxidant capacity in the aorta [94] or higher MDA in the pancreas or plasma [95,96]. Similarly, SHR rats had higher SOD activity in the heart and liver than their WKY counterparts [97]. Methyldopa administration lowered MDA in the placenta, leading to the values obtained in WKY control rats, which probably can be considered as a positive phenomenon. For SOD, a similar effect of the drug was observed and the elevation of both parameters reached values similar to the SHR control rats. The combined baicalein and methyldopa treatment led to the values observed in SHR control rats. On the other hand, the effect of the combined administration of methyldopa extract was different, as it led to the values observed for the WKY control animals. Summing up, it seems that in general, the combined administration of the substances used with methyldopa had a positive effect on the determined parameters of oxidation, which is consistent with the information on the antioxidant properties of baicalin in rats with induced hypertension [26,98].

A certain limitation of the obtained results and their interpretation was the fact that during the administration of all substances, inhibition of the increase in body weight of SHR animals was noted. 

## 4. Conclusions

To conclude, we were able to demonstrate that baicalein and the extract exhibited a broad profile of biochemical and molecular activity, with a tendency to have a more significant effect in the cells of the heart and to a lesser extent in the placenta of the studied animals. The flavonoid in co-medication with methyldopa in the applied doses was able to lower blood pressure in pregnant rats. The obtained results brought us closer to understanding the mechanism of action of baicalein, in particular in combination with the reference drug methyldopa. Such results may constitute the basis for downstream investigations in this matter, in particular with the participation of baicalein, aimed at investigating its in-depth effects, understanding its fate in the body, with particular emphasis on the combined application with methyldopa, together with a broadened knowledge of the molecular basis of their transport across the blood–placenta barrier. In the longer term of such experiments, the positive results may contribute to the development of a new antihypertensive drug for patients suffering from preeclampsia or pregnancy-induced hypertension.

## 5. Materials and Methods

### 5.1. Chemicals

Methyldopa and baicalein (purity: >99.0%) were provided by Sigma-Aldrich (Poznań, Poland).

### 5.2. Plant Material

For the extraction, roots of Baikal skullcap (*Scutellaria baicalensis* Georgi) were harvested at the beginning of October 2019 from cultivation in the Garden of Medicinal Plants in Plewiska, near Poznań (Institute of Natural Fibres and Medicinal Plants—National Research Institute, Poznań, western Poland). The collected plant material was cleaned, cut into small pieces, and dried at 45 °C with a relative humidity of 20% in a UZ-108 heating chamber (GoBest, Poznań, Poland). Voucher specimens were identified by the authors, and deposited in the herbarium of the Department of Breeding and Botany of Useful Plants (Institute of Natural Fibres and Medicinal Plants—National Research Institute).

### 5.3. Extract from Scutellaria baicalensis Roots

The procedure for obtaining the extract was similar to that in our previous work [99]. Briefly, the powered Baikal skullcap roots were extracted with purified water for 3 h at 90 °C (1:10 plant material to solvent ratio). After filtering, the extract was frozen at −55 °C and next lyophilized, then the dry plant extract was stored at 20–25 °C. The extract was prepared in the Department of Pharmacology and Phytochemistry (Institute of Natural Fibres and Medicinal Plants—National Research Institute). 

### 5.4. Determination of Bioactive Compounds in the Extract

The analysis was prepared in accordance with European Pharmacopoeia 8, monograph Baikal skullcap root. Sample preparation: Approximately 300 mg of dry extract was placed in a 100 mL round-bottomed flask and extracted with 40 mL of 70% ethanol under the reflux condenser for 3 h. After cooling down, the sample was filtered and transferred to a 100 mL volumetric flask. The filter was washed with 70% ethanol and the sample was filled up with the same solution. Next, 1 mL of the solution was transferred to a 50 mL volumetric flask and diluted to 50 mL with methanol. The solution was passed through 0.45 μm filters. Test solution: Approximately 1 mg of every standard was placed (separately) in a 10 mL volumetric flask and dissolved with methanol. Calibration curves were used to determine the concentration in the sample. The separation was performed by HPLC-DAD (Agilent 1100, Santa Clara, CA, USA) on a LiCHrosphere C18e column, 125 mm × 4 mm × 5 µm (Merck, Darmstadt, Germany). The mobile phases were as follows: 0.1% phosphoric acid in water (*V*/*V*) (phase A) and acetonitrile (phase B). The flow rate was 1.0 mL/min. The assay was performed in a gradient elution procedure as follows: 0.0 min—90% of phase A, 30 min—60% of phase A. The column temperature was 25 °C, and the detection was at ʎ = 280 nm. The peaks were identified by comparing the retention time and UV–VIS spectra with those of the standard solution.

### 5.5. Animals

Twenty-week-old female rats with persistent arterial hypertension (SHR, *n* = 40) and their analogues without hypertension (WKY, *n* = 10) were obtained from the Charles River Laboratories, Sulzfeld (Germany). Male animals (SHR, *n* = 5; WKY, *n* = 5) of the same age were ordered to induce pregnancy. The males were not the subject of experiments.

All animals were kept in groups of five rats, in plastic cages (60 (l) × 35 (w) × 20 (h) cm with stainless steel covers) at a temperature of 20 ± 2 °C, 65–75% humidity, and a reversed circadian cycle (7 p.m. to 7 a.m. light). The rats had free access to tap water and standard laboratory diet (pellets, Labofeed B, PN-ISO 9001, Żurawia, Poland).

All female animals were mated with respective male strains at estrous for three consecutive days. Pregnancy was determined by the presence of a spermatozoa plug in the vagina, noted as day 0 of pregnancy.

### 5.6. Animal Groups

All rats were assigned randomly to five groups of animals (*n* = 50). The first group was a negative control (WKY control) group (*n* = 10); rats received vehicle—0.5% methylcellulose solution in the appropriate volume. A scheme of the experiment is presented in Figure 14.

The second group was a positive control (SHR control) group (*n* = 10), which was treated analogously to the WKY control group—0.5% methylcellulose (MC) solution in the appropriate volume. The third group consisted of SHR (SHR-methyldopa) rats (*n* = 10) receiving only methyldopa (total 400 mg/kg bw/day; intragastrically = p.o.). The fourth group consisted of SHR animals (*n* = 10) receiving methyldopa + baicalein (SHR-methyldopa + baicalein) (methyldopa—400 mg/kg bw/day; p.o. + baicalein—200 mg/kg bw/day; p.o.). The last, fifth group of animals were SHR rats (*n* = 10) receiving methyldopa + *S. baicalensis* root extract (methyldopa—400 mg/kg bw/day; p.o. + extract (1000 mg/kg bw/day; p.o.) (SHR-methyldopa + extract).

The body mass of rats was measured at the beginning of the experiment, then once a week, and on the last day of the experiment.

### 5.7. Drug Treatment

After confirming pregnancy, the seventh day after fertilization, WKY and SHR rats were treated as in the scheme described above (Section 5.6), twice daily, i.e., morning (8:00 AM) and evening (8:00 PM), 1 h before the cardiovascular tests—for 14 consecutive days—for termination of pregnancy. Baicalein, extract, and methyldopa were suspended in the 0.5% MC solution. Control rats received 0.5% MC p.o. solution as a vehicle in proportion to body mass, not exceeding 5 ml in volume.

### 5.8. Cardiovascular Studies

Cardiovascular studies (evaluation of systolic and diastolic blood pressure—SBP and DBP, respectively) and monitoring of rats’ heart activity (heart rate—HR) were carried out 3 times daily using mouse–rat tail cuff plethysmography (IITC Life Science Inc., Woodland Hills, CA, USA). This experiment schedule was conducted in order to optimally observe the substance’s effects in circadian rhythm (8:00 AM, 12:00AM, and 8:00 PM) and in timescale (day 7 to 21). This observation was conducted to check the antihypertensive effect in pregnant rats (SHR) and possible side effects of the substances as well.

On day 14 of treatment (21st day of the pregnancy), one hour after administration of the compounds, the rats were decapitated. The heart and placenta of the animals were collected and stored at −80 °C. In the tissues, the levels of planned biochemical and molecular parameters were measured for the assessment of the influence of treatment with the preventive substances in the pregnant rats.

### 5.9. Biochemical Tests 

The analyses focused on the above-mentioned factors related to inflammatory processes (TNF-α, Il-1β, IL-6) and vascular diseases in human peripheral vascular and placental endothelial cells (TGF-β, HIF-1α, VEGF, PlGF) and their molecular analysis at the level of mRNA and at the protein level (amount or activity) of selected markers/enzymes of oxidative stress (superoxide dismutase (SOD) and malondialdehyde (MDA/TBARS analysis). Additionally, changes in the level of several indicators of myocardial damage (the concentration of troponin cTnC and cTnI, creatine kinase, myoglobin, and lactate dehydrogenase) were evaluated by measuring their concentration changes in the heart tissue collected from the above-mentioned rats.

#### 5.9.1. Analysis of Changes in mRNA Levels 

The isolation of total cellular RNA was performed according to the manufacturer’s protocol of TriPure Isolation Reagent (Roche, Mannheim, Germany). The synthesis of complementary DNA was performed using the Transcriptor First-Strand Synthesis System (Roche, Mannheim, Germany) according to the manufacturer’s protocol. The obtained transcripts were used directly for the quantitative real-time PCR (RT-PCR) or stored at −20 °C. The mRNA levels of the genes of interest (GOIs)—PlGF, VEGF, TNF-α, HIF-1α, TGF-β, IL-1β, and IL-6—were analyzed by real-time quantitative PCR technique using a LightCycler96 Instrument (Roche, Mannheim, Germany) and a LightCycler480 Probes Master kit (Roche, Mannheim, Germany). The PCR program was initiated with activation at 95 °C for 10 min. Each PCR cycle comprised a denaturation step at 95 °C, an annealing step at a specific temperature, and an extension step at 72 °C. The increase in fluorescence of each PCR product during the amplification was measured and the data were visualized as an amplification curve using the implemented LightCycler96 software Version 1.1. The sequences of primers were designed using the Oligo 6.0 program (National Biosciences, Colorado Springs, CO, USA). A strategy of relative quantification of gene expression was applied; therefore, for that purpose, the GAPDH (glyceraldehyde 3-phosphate dehydrogenase) gene was used as a housekeeping gene for normalization of fluorescence change data obtained during the PCR amplifications of GOIs. All oligonucleotide sequences were synthesized by Environmental DNA Sequencing Laboratory Oligo, Institute of Biochemistry and Biophysics (Warsaw, Poland). Their nucleotides composition and amplicon lengths are summarized in Table 4.

#### 5.9.2. Biochemical Parameters 

The activity of antioxidant enzymes was investigated in supernatants from tissue homogenates using colorimetric methods based on changes in the absorbance wavelength of the reaction products. 

##### SOD

To determine SOD activity, the test tissues were homogenized in HEPES buffer (pH 7.2) containing 1 mM EGTA, 210 mM mannitol, and 70 mM sucrose per gram of tissue. The homogenates were then centrifuged at 1500× *g* for 5 min at 4 °C. For the analysis, the supernatants from above tissue homogenates and a commercial reaction kit—Superoxide Dismutase Assay Kit (No. 706002) (Cayman Chemical, Ann Arbor, Michigan, USA)—were used, and an analytical assumption was based on the phenomenon of reduction of nitrotetrazolium salt of superoxide radicals generated by xanthine oxidase and hypoxanthine. One SOD enzyme unit was defined as the amount of enzyme needed to inhibit the formazan reaction by 50%. Absorbance was measured using an EPOCH spectrophotometer (Biotek, Santa Clara, CA, USA) at a wavelength of 450 nm. SOD activity was calculated from the standard curve. Serum SOD was analyzed analogously; samples were diluted 1:5 according to the manufacturer’s protocol. 

##### MDA

Tissue lipid peroxidation was also measured by an indirect method by determining the level of malonyl dialdehyde (MDA). The analytical assumption was based on the knowledge that products of peroxidation of polyunsaturated fatty acids form colored complexes with thiobarbituric acid (TBARS) to form MDA. The commercial TBARS (TCA Method) Assay Kit (Cayman, No. 700870) (Cayman Chemical, Ann Arbor, MI, USA) was applied for the tests. Briefly, tissues were homogenized in RIPA buffer (Sigma, No. R0278) (Burlington, MA, USA), then centrifuged at 1600× *g* for 10 min at 4 °C. The colorimetric test was carried out according to the manufacturer of the kit. The resulting MDA–TBA complexes were measured at a wavelength of 535 nm and the MDA concentration in µM was calculated from the curve against the standard. 

#### 5.9.3. Myocardial Proteins

Myocardial proteins were measured by enzyme-linked immunosorbent assay (ELISA) using target-specific, commercially available, ready-made kits from EIAAB Science (Wuhan, China): Rat Myoglobin ELISA Kit (Cat. No. E0480r), Rat Troponin I, cardiac muscle ELISA Kit (Cat. No. E0478r), Rat Troponin T, cardiac muscle ELISA Kit (Cat. No. E1339r), Rat Creatine kinase M-type ELISA Kit (Cat. No. E0479r), Rat Creatine kinase B-type ELISA Kit (Cat. No. E2030r), Rat L -Lactate dehydrogenase A chain, LDH-A ELISA Kit (Cat. No. E2195r). 

Homogenate (10%) in the PBS buffer was centrifuged for 5 min at 5000× *g*. The supernatant was used for the tests according to the manufacturer’s instructions. Absorbance reading was performed at 450 nm along with the standard. Protein concentration in the samples was calculated based on the analysis of the fit curves and data performed in Curve Expert 1.38 software.

### 5.10. Statistical Analysis

All values were expressed as mean ± SEM. The statistical comparison of results was carried out using one-way analysis of variance (ANOVA I) or two-way analysis of variance with replication (ANOVA II) followed by the Newman–Keuls test as a post-hoc test for detailed data analysis. The values of *p* < 0.05 were considered to indicate a statistically significant difference. All analyses were calculated using STATISTICA 13.0 software.

## Figures and Tables

**Figure 1 pharmaceuticals-15-01342-f001:**
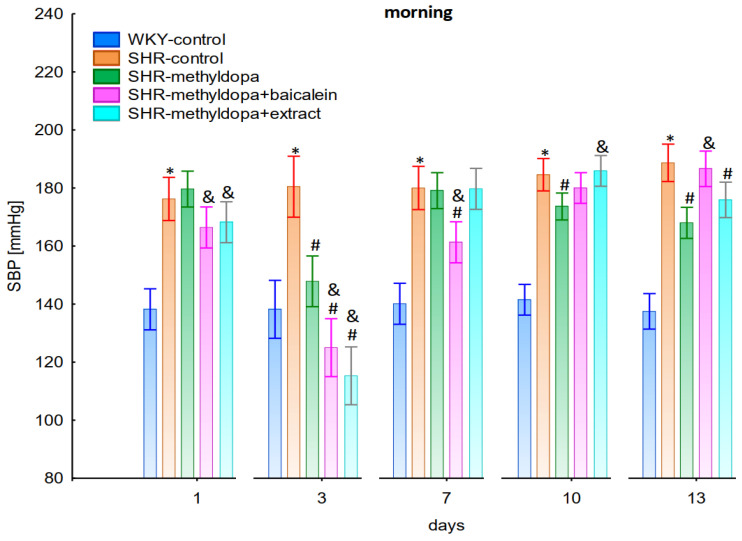
Influence of methyldopa and its combination with baicalein or extract from *Scutellaria baicalensis* roots on morning systolic blood pressure (SBP) in SHR pregnant rats. Legend: *n* = 10, mean ± SEM; *—vs. WKY control, *p* < 0.05; #—vs. SHR control, *p* < 0.05; &—vs. SHR-methyldopa, *p* < 0.05.

**Figure 2 pharmaceuticals-15-01342-f002:**
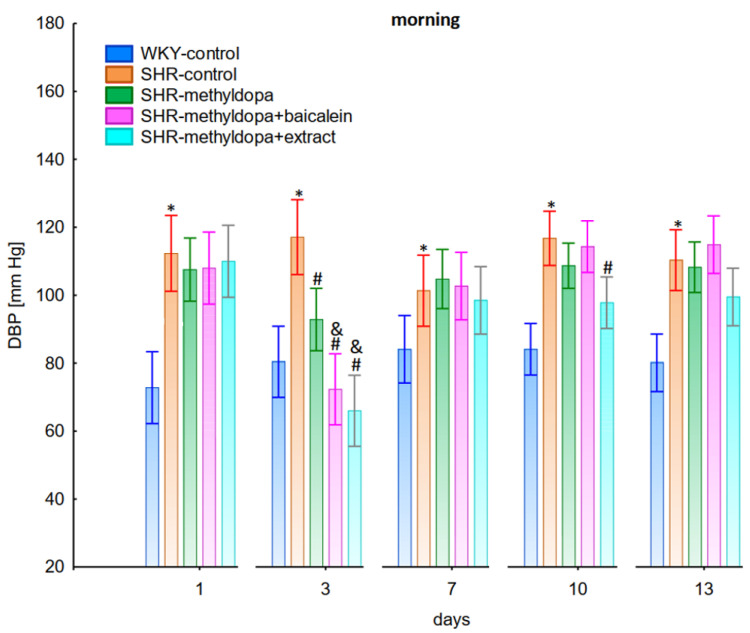
Influence of methyldopa and its combination with baicalein or extract from *Scutellaria baicalensis* roots on morning diastolic blood pressure (DBP) in SHR pregnant rats. Legend: *n* = 10, mean ± SEM; *—vs. WKY control, *p* < 0.05, #—vs. SHR control, *p* < 0.05; &—vs. SHR-methyldopa, *p* < 0.05.

**Figure 3 pharmaceuticals-15-01342-f003:**
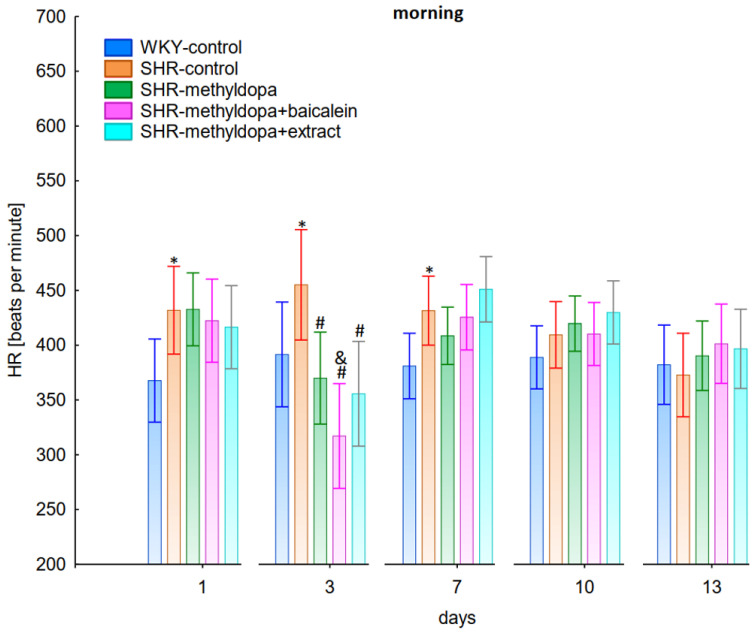
Influence of methyldopa and its combination with baicalein or extract from *Scutellaria baicalensis* roots on morning heart rate (HR) in SHR pregnant rats. Legend: *n* = 10, mean ± SEM; *—vs. WKY control, *p* < 0.05; #—vs. SHR control, *p* < 0.05; &—vs. SHR-methyldopa, *p* < 0.05.

**Figure 4 pharmaceuticals-15-01342-f004:**
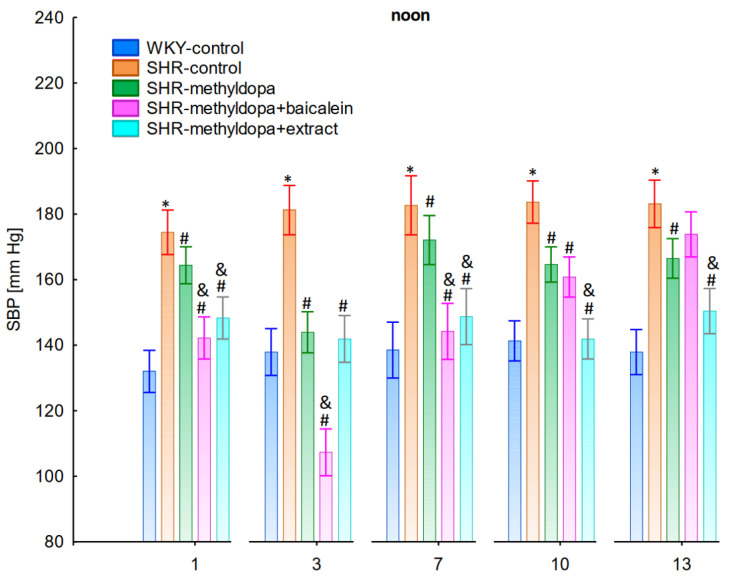
Influence of methyldopa and its combination with baicalein or extract from *Scutellaria baicalensis* roots on noon systolic blood pressure (SBP) in SHR pregnant rats. Legend: *n* = 10, mean ± SEM; *—vs. WKY control, *p* < 0.05; #—vs. SHR control, *p* < 0.05; &—vs. SHR-methyldopa, *p* < 0.05.

**Figure 5 pharmaceuticals-15-01342-f005:**
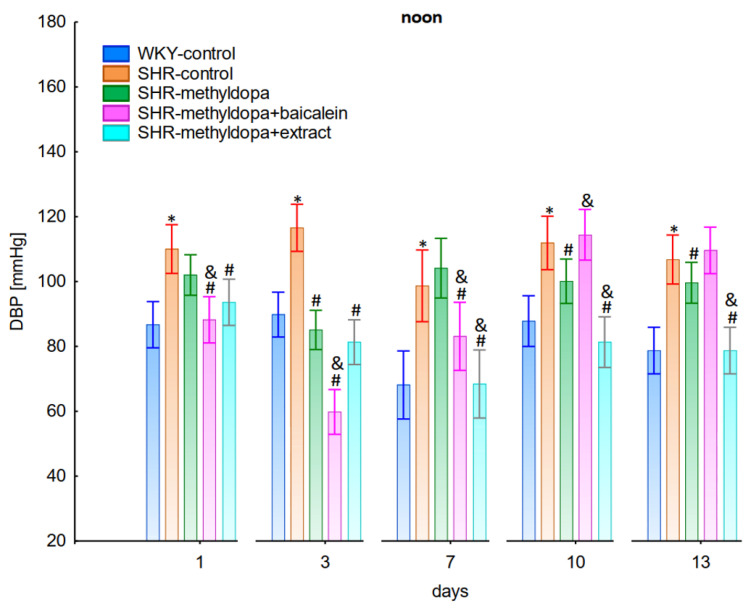
Influence of methyldopa and its combination with baicalein or extract from *Scutellaria baicalensis* roots on noon diastolic blood pressure (DBP) in SHR pregnant rats. Legend: *n* = 10, mean ± SEM; *—vs. WKY control, *p* < 0.05; #—vs. SHR control, *p* < 0.05; &—vs. SHR-methyldopa, *p* < 0.05.

**Figure 6 pharmaceuticals-15-01342-f006:**
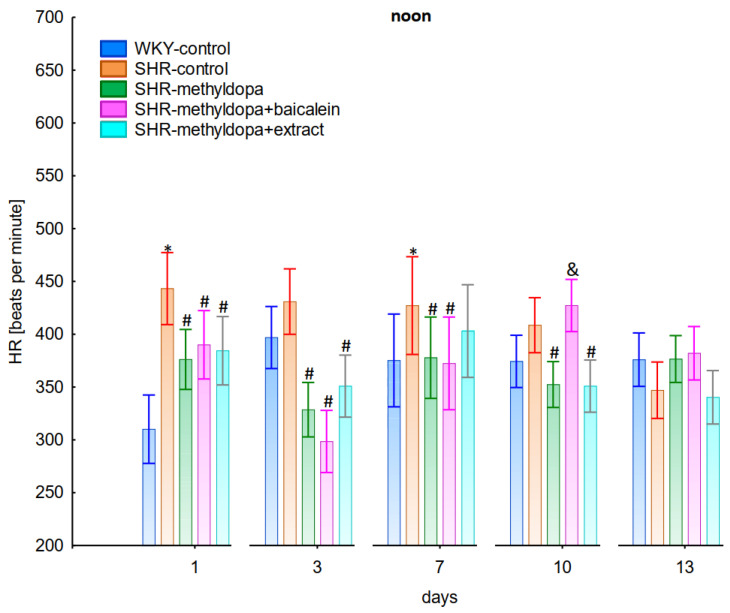
Influence of methyldopa and its combination with baicalein or extract from *Scutellaria baicalensis* roots on noon heart rate (HR) in SHR pregnant rats. Legend: *n* = 10, mean ± SEM; *—vs. WKY control, *p* < 0.05; #—vs. SHR control, *p* < 0.05; &—vs. SHR-methyldopa, *p* < 0.05.

**Figure 7 pharmaceuticals-15-01342-f007:**
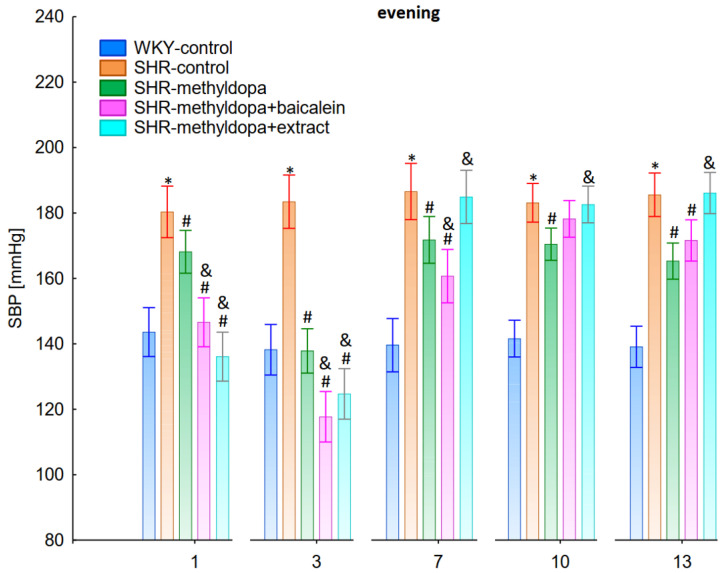
Influence of methyldopa and its combination with baicalein or extract from *Scutellaria baicalensis* roots on evening systolic blood pressure (SBP) in SHR pregnant rats. Legend: *n* = 10, mean ± SEM; *—vs. WKY control, *p* < 0.05; #—vs. SHR control, *p* < 0.05; &—vs. SHR-methyldopa, *p* < 0.05.

**Figure 8 pharmaceuticals-15-01342-f008:**
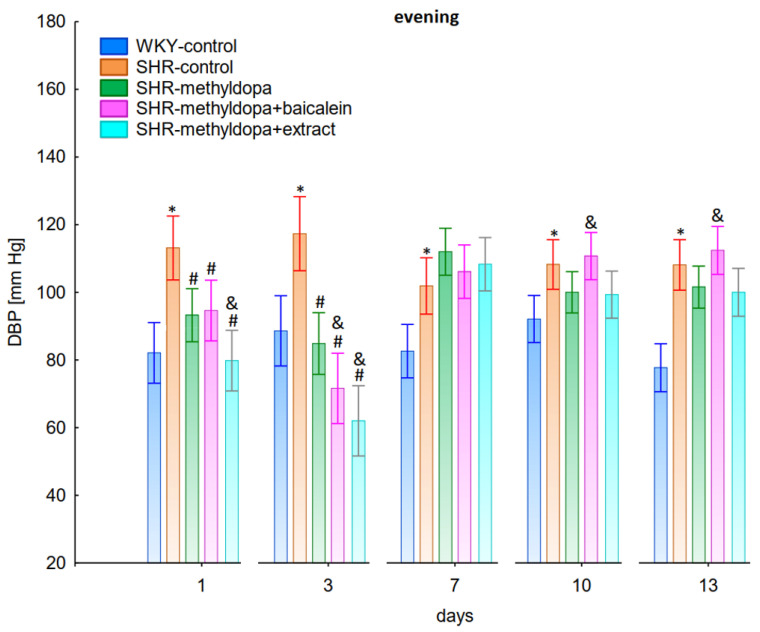
Influence of methyldopa and its combination with baicalein or extract from *Scutellaria baicalensis* roots on evening diastolic blood pressure (DBP) in SHR pregnant rats. Legend: *n* = 10, mean ± SEM; *—vs. WKY control, *p* < 0.05; #—vs. SHR control, *p* < 0.05; &—vs. SHR-methyldopa, *p* < 0.05.

**Figure 9 pharmaceuticals-15-01342-f009:**
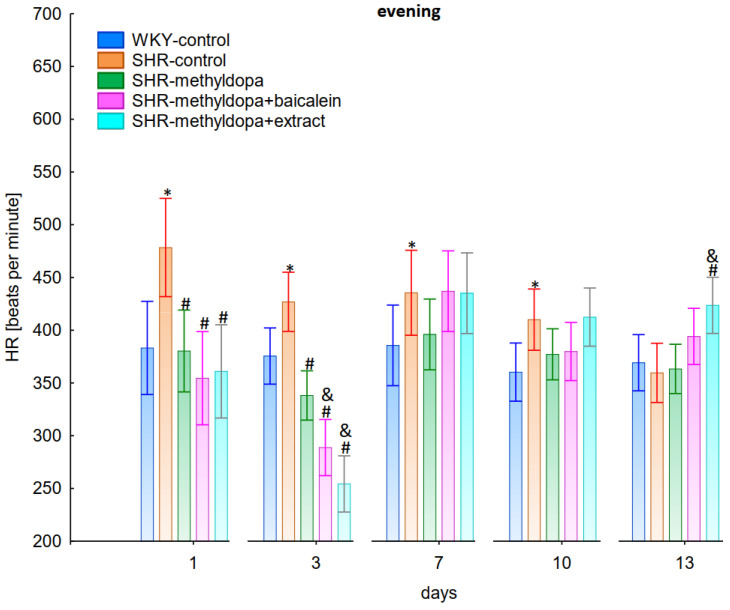
Influence of methyldopa and its combination with baicalein or extract from *Scutellaria baicalensis* roots on evening heart rate (HR) in SHR pregnant rats. Legend: *n* = 10, mean ± SEM; *—vs. WKY control, *p* < 0.05; #—vs. SHR control, *p* < 0.05; &—vs. SHR-methyldopa, *p* < 0.05.

**Figure 10 pharmaceuticals-15-01342-f010:**
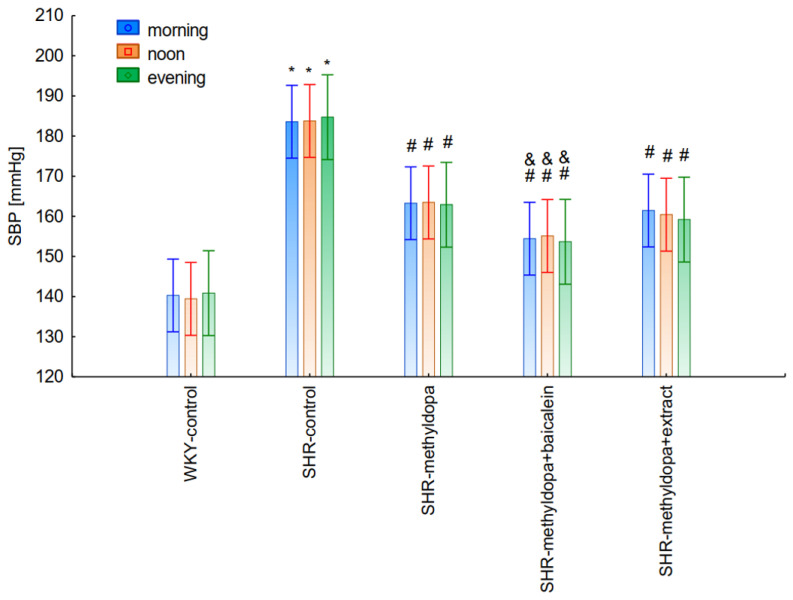
Changes in systolic blood pressure (SBP) after administration of methyldopa and its combination with baicalein and *Scutellaria baicalensis* root extract with regard to the time of day of administration in SHR pregnant rats. Legend: *n* = 10, mean ± SEM; *—vs. proper WKY control, *p* < 0.05; #—vs. proper SHR control, *p* < 0.05; &—vs. proper SHR-methyldopa, *p* < 0.05.

**Figure 11 pharmaceuticals-15-01342-f011:**
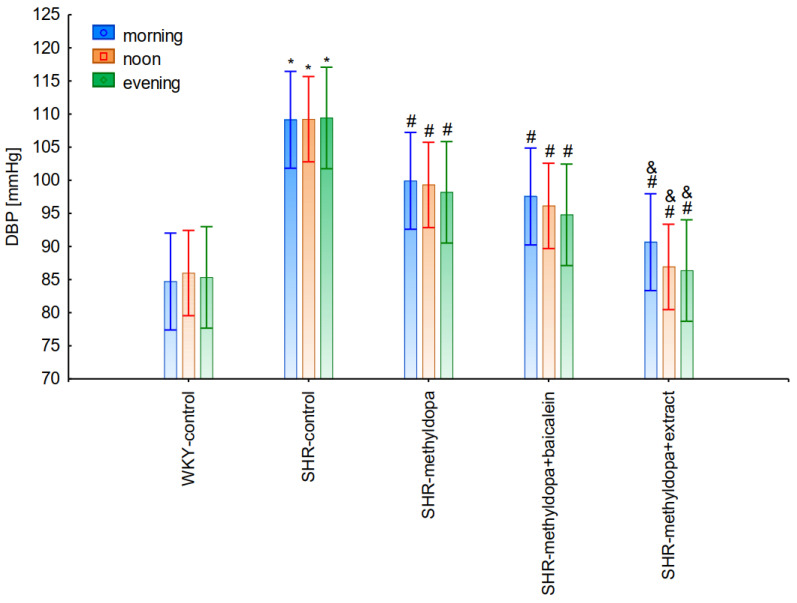
Changes in diastolic blood pressure (DBP) after administration of methyldopa and its combination with baicalein and *Scutellaria baicalensis* root extract with regard to the time of day of administration in SHR pregnant rats. Legend: *n* = 10, mean ± SEM; *—vs. proper WKY control, *p* < 0.05; #—vs. proper SHR control, *p* < 0.05; &—vs. proper SHR-methyldopa, *p* < 0.05.

**Figure 12 pharmaceuticals-15-01342-f012:**
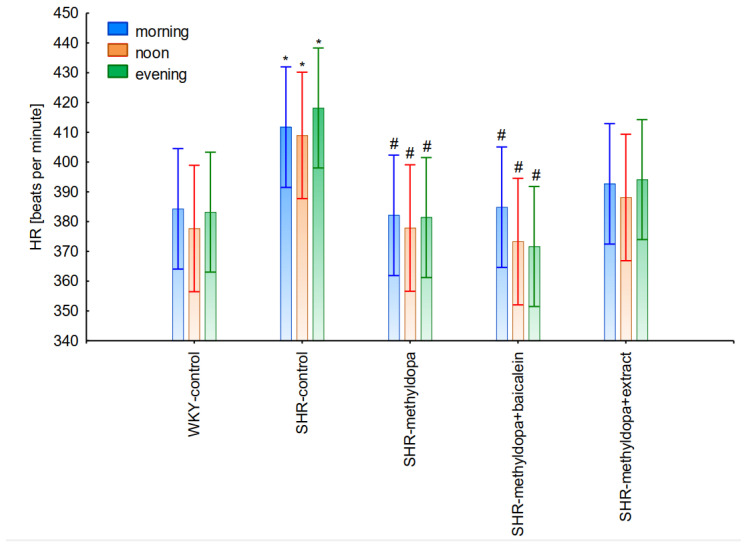
Changes in heart rate (HR) after administration of methyldopa and its combination with baicalein and *Scutellaria baicalensis* root extract with regard to the time of day of administration in SHR pregnant rats. Legend: *n* = 10, mean ± SEM; *—vs. proper WKY control, *p* < 0.05; #—vs. proper SHR control, *p* < 0.05.

**Figure 13 pharmaceuticals-15-01342-f013:**
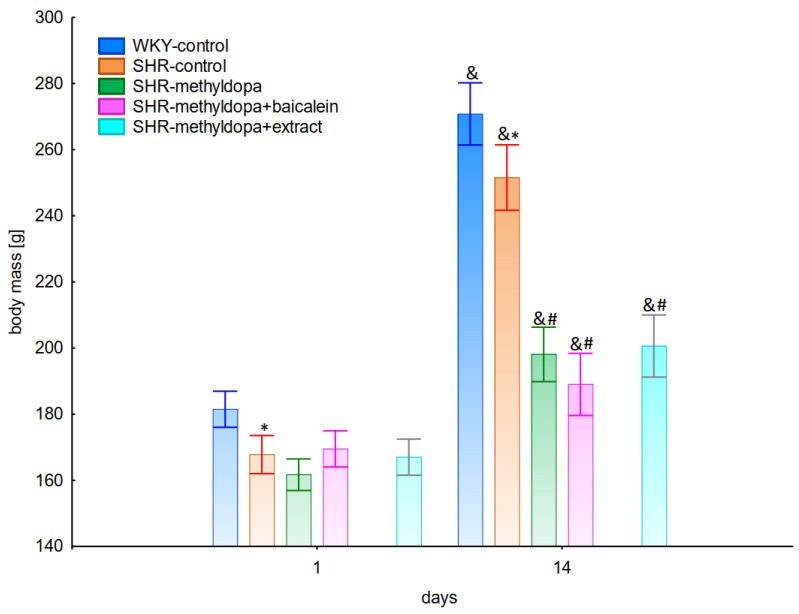
Influence of methyldopa and its combination with baicalein or extract from *Scutellaria baicalensis* roots on body mass in SHR pregnant rats. Legend: *n* = 10, mean ± SEM; *—vs. WKY control, *p* < 0.05; #—vs. SHR control, *p* < 0.05; &—vs. proper the 1st day, *p* < 0.05.

**Figure 14 pharmaceuticals-15-01342-f014:**
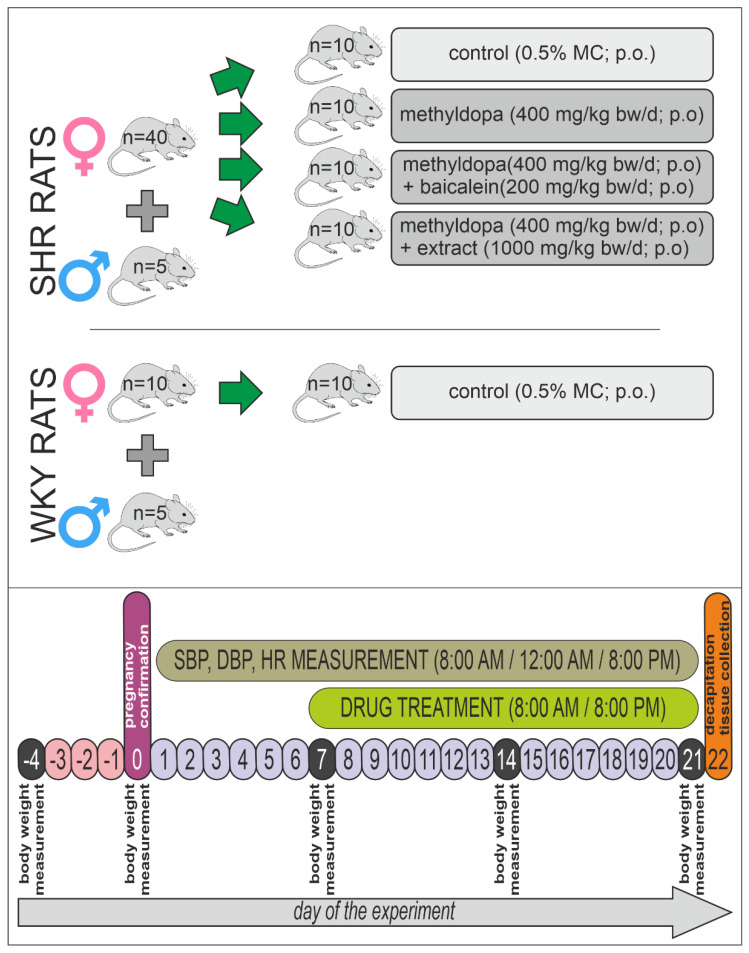
Graphical presentation of the animal experiment scheme.

**Table 1 pharmaceuticals-15-01342-t001:** Influence of methyldopa, its combination with baicalein or extract from *Scutellaria baicalensis* roots on mRNA level changes of cardiovascular factors (VEGF, HIF-1α, TGF-β, PlGF) and inflammatory processes (TNF-α, IL-1β, IL-6) in tissues of SHR pregnant rats. The numerical values represent the changes in the level of fluorescence for individual genes measured in the qPCR reaction, where the value 1—was the reference level for the referential cDNA.

Group	VEGF	HIF-1α	TGF-β	PlGF	TNF-α	IL-1β	IL-6
	Heart						
WKY control	1.00 ± 0.02	1.00 ± 0.02	1.00 ± 0.01	1.00 ± 0.03	1.00 ± 0.02	1.00 ± 0.02	1.00 ± 0.02
SHR control	0.93 ± 0.02 *	0.91 ± 0.03 *	0.88 ± 0.02 *	1.10 ± 0.03 *	0.97 ± 0.01	0.92 ± 0.02 *	1.05 ± 0.02
SHR-methyldopa	0.89 ± 0.03	0.88 ± 0.03	0.86 ± 0.02	0.99 ± 0.02 ^#^	0.95 ± 0.02	0.83 ± 0.02 ^#^	1.11 ± 0.03
SHR-methyldopa + baicalein	0.99 ± 0.01 ^&^	0.86 ± 0.02	0.83 ± 0.02	1.11 ± 0.03 ^&^	0.96 ± 0.03	0.89 ± 0.03 ^&^	1.08 ± 0.03
SHR-methyldopa + extract	0.84 ± 0.02 ^#^	0.85 ± 0.05	0.87 ± 0.02	1.11 ± 0.03 ^&^	0.98 ± 0.03	0.87 ± 0.01	1.04 ± 0.02
	Placenta						
WKY control	0.57 ± 0.01	0.58 ± 0.03	0.69 ± 0.02	1.04 ± 0.03	0.82 ± 0.02	0.79 ± 0.02	1.03 ± 0.03
SHR control	0.77 ± 0.03 *	0.78 ± 0.04 *	0.82 ± 0.04 *	1.08 ± 0.03	0.88 ± 0.03	0.80 ± 0.04	1.07 ± 0.02
SHR-methyldopa	0.64 ± 0.01 ^#^	0.60 ± 0.01 ^#^	0.71 ± 0.01 ^#^	1.06 ± 0.03	0.84 ± 0.01	0.75 ± 0.01	1.04 ± 0.04
SHR-methyldopa + baicalein	0.75 ± 0.02 ^&^	0.69 ± 0.01 ^&^	0.85 ± 0.01 ^&^	1.02 ± 0.02	0.89 ± 0.05	0.77 ± 0.06	1.05 ± 0.08
SHR-methyldopa + extract	0.65 ± 0.04 ^#^	0.64 ± 0.03 ^#^	0.70 ± 0.01 ^#^	1.16 ± 0.06	0.83 ± 0.01	0.72 ± 0.03	0.98 ± 0.05

Legend: mean ± SEM; *n* = 10; *—vs. WKY control, *p* < 0.05; #—vs. SHR control, *p* < 0.05; &—vs. SHR methyldopa, *p* < 0.05.

**Table 2 pharmaceuticals-15-01342-t002:** Influence of methyldopa and its combination with baicalein or extract from *Scutellaria baicalensis* roots on levels of selected factors related to heart damage parameters (level of: creatine kinases (B, M), myoglobin, troponins (cTnC, cTnI), lactate dehydrogenase) in SHR pregnant rats.

Group	CKB [ng/mL]	CKM [U/mL]	Myoglobin [mg/mL]	cTnT [ng/mL]	cTnI [mg/mL]	LDH-A [U/L]
	Heart					
WKY control	352 ± 32	4070 ± 574	268 ± 74	26.6 ± 3.3	8.20 ± 0.82	32,420 ± 3203
SHR control	323 ± 57	3413 ± 580	427 ± 114 *	25.4 ± 3.4	8.31 ± 1.03	38,323 ± 5006
SHR-methyldopa	245 ± 29 ^#^	2701 ± 374	97 ± 30 ^#^	16.9 ± 2.0 ^#^	5.25 ± 0.70 ^#^	21,124 ± 3243 ^#^
SHR-methyldopa + baicalein	187 ± 19 ^# &^	2321 ± 347	134 ± 39 ^#^	19.0 ± 1.9	6.25 ± 0.56 ^#^	28,826 ± 5669
SHR-methyldopa + extract	445 ± 57 * ^# &^	4932 ± 496 ^# &^	299 ± 63 ^# &^	21.9 ± 2.4	6.28 ± 0.48 ^#^	27,746 ± 2917

Legend: *n* = 10; *—vs. WKY control, *p* < 0.05; #—vs. SHR control, *p* < 0.05; &—vs. SHR methyldopa, *p* < 0.05.

**Table 3 pharmaceuticals-15-01342-t003:** Influence of methyldopa and its combination with baicalein or extract from *Scutellaria baicalensis* roots on levels of selected factors related to oxidative stress (malonyldialdehyde concentration and activity of superoxide dismutase) in placenta of SHR pregnant rats.

Group	MDA [μM]	SOD [U/mL]
	Placenta	
WKY control	3.55 ± 0.16	1.79 ± 0.14
SHR control	4.43 ± 0.23 *	2.71 ± 0.22 *
SHR-methyldopa	2.27 ± 0.39 ^#^	2.08 ± 0.13 ^#^
SHR-methyldopa + baicalein	6.27 ± 0.84 ^# &^	2.93 ± 0.15 ^&^
SHR-methyldopa + extract	3.36 ± 0.74 ^# &^	2.06 ± 0.01 ^#^

Legend: *n* = 8; *—vs. WKY control, *p* < 0.05; #—vs. SHR control, *p* < 0.05; &—vs. SHR methyldopa, *p* < 0.05.

**Table 4 pharmaceuticals-15-01342-t004:** Sequences of primers used for RT-PCR analysis.

Gene	Primer Sequence Forward (5′→3′)	Primer Sequence Reverse (5′→3′)	bp
TNF-α	TGC TTG TTC CTC AGC CTC TT	TGA GGT ACA GGC CCT CTG AT	218
IL-1β	CGA TGC ACC TGT ACG ATC AC	TCT TTC AAC ACG CAG GAC AG	226
HIF-1α	TTG CCT TTC CTT CTC TTC TCC	CAA TCC AAG GTT GCC AAG TT	164
VEGF	CCT TGC TGC TCT ACC TCC AC	ATC CAC CCC AAA ACT TTT CC	236
TGF-β	ACA TTG ACT TCC GCA AGG AC	CCG GGT TAT GCT GGT TGT A	150
PlGF	GTT CAG CCC ATC CTG TGT CT	AGC AGG GAA ACA GTT GGC TA	244
IL-6	TGC GTC CGT AGT TTC CTT CT	GGA ATC TTC TCC TGG GG GTA	211
GAPDH	GAT GGT GAA GGT CGG TGT G	ATG AAG GGG TCG TTG ATG G	108

GAPDH—glyceraldehyde 3-phosphate dehydrogenase, reference gene for normalization.

## Data Availability

Data is contained within the article.

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
