# Peer review of "Combined Effects of Methyldopa and Baicalein or Scutellaria baicalensis Roots Extract on Blood Pressure, Heart Rate, and Expression of Inflammatory and Vascular Disease-Related Factors in Spontaneously Hypertensive Pregnant Rats"

_pharmaceuticals, 2022, doi:10.3390/ph15111342_

Round 1

Reviewer 1 Report

In their work, the authors investigated the effect of baicalein or Scutellaria baicalensis root extract interaction with methyldopa in pregnant spontaneously hypertensive rats. They demonstrated that baicalein and the extract significantly affected the heart cells and slightly less the placenta of animals. Analyzed flavonoid administered with methyldopa lowered the blood pressure in pregnant rats stronger than methyldopa alone, revealing the tested compound's previously unknown mechanism of action. Obtained results are promising; thus, they might contribute to developing preclinical studies to help patients suffering from preeclampsia or pregnancy-induced hypertension. This is a very important and significant conclusion from the research.
Interestingly and intriguingly, the flavonoid acted slightly differently than the extract containing it, sometimes in the opposite way, which provides an interesting basis for further research. All the results were collected and presented clearly using correct statistical tests, especially from systolic and diastolic blood pressure. However, I would like the authors to make minor linguistic and grammatical corrections, which caught my eye, especially in the introduction. This is the only critical note I have for the author, besides that, I am very pleased to recommend the work for publication in Pharmaceuticals.

Author Response

We are grateful for the Reviewers' remarks that contributed to improving our work. Below are the replies to the comments. We have included them together because some of the remarks are common or complementary.

All corrections in work are marked in red.

Reviewer 1

  1. I would like the authors to make minor linguistic and grammatical corrections, which caught my eye, especially in the introduction.

We thank the reviewer for this helpful comment. We have striven to improve the quality of the English language in our manuscript, taking into particular consideration the minor corrections suggested by the reviewer.

Reviewer 2

  1. However, the manuscript turns out to be too long and cumbersome, unclear with many repeated sentences almost copied. Many chapters can be merge, i.e. the three cardiovascular measurements carried out (SBP, DBP, HR) provide almost overlapping results thus they could be reported as a single chapter.

As suggested, the results for SBP, DBP and HR have been transformed into 4 new sections in which the exact data for a given time of day have been combined. However, in the case of the remaining data, we believe that their combination would be detrimental to the clarity of the message.

  1. The same goes for inflammation, vascular and myocardial damage factors and enzymes, as well markers of oxidative stress.

In the case of the remaining data, we believe that text shortening would be detrimental to the clarity of the message.

  1. I would suggest to lighten, while reducing, introduction and discussion sections.

In response to the suggestions, the indicated sections have been substantively edited and supplemented with comments from other reviewers. Hence, in our opinion, the current manuscript should be satisfactory for both Reviewers 2 and 4.

Reviewer 3

  1. On what basis, the authors choose the doses of methyldopa, baicalein and extract in the current study?

Details on dose selection for methyldopa, baicalein, and extract have been listed in the second paragraph of the discussion.

  1. The authors should add a conclusion that contains all the outlines of the study.

Conclusions have been added.

  1. Graphical abstract for the study is highly recommended.

We thank the reviewer for this helpful remark. Indeed, a graphical explanation of the experiment will facilitate understanding of the course of the experiment. We decided to add this as Figure 14 in the Materials and Methods section.

Reviewer 4

  1. Please describe more detailed, in introduction, the metyldopa action mechanism.

We would like to thank the reviewer for pointing this out. The information on methyldopa has been updated accordingly in the introduction.

  1. Please offer the mechanism of VGEF increasement in eclampsia in discussions, in order to be more clease why you asses this parameter. Eventually offer details about placenta hypoxia associated with eclampsia.

We would like to thank the reviewer for pointing this out. The information on VEGF has been updated accordingly in the discussion.

  1. Conclusions - please make a distinct chapter with conclusions.

We have implemented this suggestion.

Reviewer 2 Report

The project is interesting and potentially opens up a field of application that could play a role in medical practice.

However, the manuscript turns out to be too long and cumbersome, unclear with many repeated sentences almost copied. Many chapters can be merge, i.e. the three cardiovascular measurements carried out (SBP, DBP, HR) provide almost overlapping results thus they could be reported as a single chapter. The same goes for inflammation, vascular and myocardial damage factors and enzymes, as well markers of oxidative stress.

I would suggest to lighten, while reducing, introduction and discussion sections. 

Author Response

(The authors gave the same response as above.)

Reviewer 3 Report

The current manuscript concerns with the combined effects of methyldopa, baicalein or extract from roots of Scutellaria baicalensis on the blood pressure, heart rate, expression of inflammatory factors in SHR pregnant rats. It is well-planned study and the work is novel, interesting and can be considered for publication after minor revision.

Suggesting the authors to address these issues:

1- On what basis, the authors choose the doses of methyldopa, baicalein and extract in the current study?

2- The authors should add a conclusion that contains all the outlines of the study.

3-    Graphical abstract for the study is highly recommended.

Author Response

(The authors gave the same response as above.)

Reviewer 4 Report

The manuscript entitled:
"Combined effects of methyldopa and baicalein or extract from 2
roots of Scutellaria baicalensis on the blood pressure, heart rate, 3
expression of selected factors related to inflammatory processes, 4
and vascular diseases in SHR pregnant rats" brings a new inside in eclampsia treatment, using nutraceuticals as associated therapy.

The following observations have to be made:

Please describe more detailed, in introduction, the metyldopa action mechanism.

Please offer the mechanism of VGEF increasement in eclampsia in discussions, in order to be more clease why you asses this parameter. Eventually offer details about placenta hypoxia associated with eclampsia.

Conclusions - please make a distinct chapter with conclusions.

Author Response

(The authors gave the same response as above.)

Round 2

Reviewer 2 Report

The manuscript, in my opinion, is still too long but it can be accepted.